# Reviewing the Limitations of Adult Mammalian Cardiac Regeneration: Noncoding RNAs as Regulators of Cardiomyogenesis

**DOI:** 10.3390/biom10020262

**Published:** 2020-02-10

**Authors:** Robin Verjans, Marc van Bilsen, Blanche Schroen

**Affiliations:** 1Department of Cardiology, Cardiovascular Research Institute Maastricht (CARIM), Maastricht University, 6200 MD Maastricht, The Netherlands; r.verjans@maastrichtuniversity.nl; 2Department of Physiology, Cardiovascular Research Institute Maastricht (CARIM), Maastricht University, 6200 MD Maastricht, The Netherlands; marc.vanbilsen@maastrichtuniversity.nl

**Keywords:** cardiac regeneration, cardiomyogenesis, heart failure, long noncoding RNA, microRNA

## Abstract

The adult mammalian heart is incapable of regeneration following cardiac injury, leading to a decline in function and eventually heart failure. One of the most evident barriers limiting cardiac regeneration is the inability of cardiomyocytes to divide. It has recently become clear that the mammalian heart undergoes limited cardiomyocyte self-renewal throughout life and is even capable of modest regeneration early after birth. These exciting findings have awakened the goal to promote cardiomyogenesis of the human heart to repair cardiac injury or treat heart failure. We are still far from understanding why adult mammalian cardiomyocytes possess only a limited capacity to proliferate. Identifying the key regulators may help to progress towards such revolutionary therapy. Specific noncoding RNAs control cardiomyocyte division, including well explored microRNAs and more recently emerged long noncoding RNAs. Elucidating their function and molecular mechanisms during cardiomyogenesis is a prerequisite to advance towards therapeutic options for cardiac regeneration. In this review, we present an overview of the molecular basis of cardiac regeneration and describe current evidence implicating microRNAs and long noncoding RNAs in this process. Current limitations and future opportunities regarding how these regulatory mechanisms can be harnessed to study myocardial regeneration will be addressed.

## 1. Introduction

The heart is one of the most vital organs of the human body but at the same time forms one of the least adaptive organs in terms of regenerative capacity. The human adult myocardium, like that of most mammalians, is incapable of adequate regeneration following cardiac injury. This failure to repair severe cardiomyocyte loss is a leading cause of heart failure and death worldwide [1]. One of the most prominent barriers limiting cardiac regeneration is the inability of the majority of adult cardiomyocytes to proliferate. However, the traditional belief that the mammalian heart is a permanent post-mitotic organ has recently been overturned. It has become clear that the human heart is capable of modest regeneration. Cardiomyocyte turnover frequencies have been reported varying from 1% to 22% annually [2,3]. This cellular turnover is realized, at least partly, by pre-existing cardiomyocytes undergoing cell division [2,3]. Importantly, studies indicate that mammals even have the capability to overcome severe cardiomyocyte loss during neonatal life [4]. These exciting findings gave rise to the idea that it might be possible to stimulate cardiac regeneration through cardiomyocyte proliferation in order to repair myocardial injury or treat failure. Understanding the restrictions of adult mammalian cardiomyocytes to proliferate and identifying its regulators is crucial to achieve this ambitious therapeutic goal. 

An important group of regulators are the noncoding RNAs (ncRNAs). These molecules are involved in the regulation of practically all biological processes, particularly during development and disease [5]. Different classes of ncRNAs have been linked to cardiac regeneration in which microRNAs (miRNAs) and long noncoding RNAs (lncRNAs) are among the best explored. Over the past two decades, various conserved miRNA families and clusters have been investigated for their ability to increase cardiomyogenesis [6,7]. Although miRNA research had a head start, evidence is accumulating that the more recently emerged lncRNAs are also critically involved in cardiomyocyte proliferation [8]. Gain- and loss-of-function studies show that modulation of specific miRNAs and lncRNAs holds promise to promote the regenerative capacity of the mammalian heart in small [9] and large [10] animal models. However, further identification of ncRNA candidates and elucidation of the gene networks controlling cardiac regeneration are required. The role of ncRNA classes other than miRNAs and lncRNAs, such as circular RNAs, transfer RNAs (fragments), and small nuclear RNAs, in controlling cardiomyogenesis has not yet been extensively studied. Therefore, this review highlights the molecular basis of cardiac regeneration among species and describes the miRNAs and lncRNAs controlling them. Current limitations and future opportunities how these regulatory mechanisms can be harnessed to study myocardial regeneration will be addressed.

## 2. Cardiac Regeneration and Cardiomyocyte Renewal Among Species—Lessons Learned from Evolution

Cardiac regenerative potential has been compared among vertebrate species with the aim to find a molecular explanation for the limitations in human cardiomyogenesis. In the following paragraphs, we describe the regenerative capacity of the newt, zebrafish, mouse, and human heart and describe similarities and differences between them.

### 2.1. Lower Vertebrates Can Regenerate Their Myocardium throughout Life

Lower vertebrates such as the newt [11,12] and zebrafish [13,14,15], have an astonishing ability to replace lost cardiac tissue by proliferation of pre-existing cardiomyocytes. Availability of genetic and molecular tools has made the zebrafish the best characterized heart regeneration model to date. Zebrafish have a two-chambered heart that pumps blood to the body and the gills and can fully regenerate a cardiac amputation of 20% within 2 months [13,16,17]. Different myocardial regeneration models have been designed in which myocardial injury is induced by ventricular apical resection [18], genetic ablation [19], or cryoinjury [20,21,22]. Robust myocardial regeneration is observed in all models, although the dynamics of the regenerative process may differ due to variations in the size of injured myocardium and the loss of other cell types. Defining similarities and differences (see paragraph 2.4) between the mechanisms underlying cardiomyocyte proliferation between lower vertebrates and mammals may increase our understanding as to how cardiomyocyte proliferation can be induced in humans.

### 2.2. Murine Cardiomyogenesis Is Limited by Time and Number

In contrast to lower vertebrates, mammalian species such as mice have a limited capacity for cardiomyocyte division. Studies using multi-isotope imaging mass spectrometry to study cardiomyocyte turnover during normal ageing show that cardiomyocyte renewal occurs at a slow annual rate of 0.76% in mice [3]. Importantly, these newly formed cardiomyocytes are shown to derive from the division of pre-existing cardiomyocytes [3]. Furthermore, neonatal mice possess the ability to regenerate their myocardium in response to a range of injury models similar to models applied in zebrafish, including ventricular resection [4,23], myocardial infarction [24], cryo-injury [25,26], and clamping [27]. Although the inflicted extent and applied mode of injury in the model can affect the regenerative response [27], all models have shown that the major source of cardiomyocyte renewal are pre-existing cardiomyocytes that re-enter the cell cycle [24], as in lower vertebrates. Importantly, while 1-day old mouse pups are able to undergo full regeneration in response to injury, 7-days old pups are not [4]. These studies identify a specific time window of the first week following birth in which the murine heart is still capable of adequate regeneration, after which this capability is lost. 

Alkass and colleagues [28] monitored cardiomyocyte proliferation, multinucleation, and polyploidization over time during murine postnatal development. They showed that the majority of cardiomyocytes are set within the first 11 postnatal days using stereological analysis. The number of cardiomyocytes expanded from ~1.7 million at postnatal day 2 until reaching a plateau of 2.6 million at postnatal day 11, which remained constant at least until p100 [28]. This event is subsequently followed by two waves of DNA synthesis without cytokinesis during the second and third postnatal weeks, which renders most of them binucleated [28]. At this point, cell division ceases and postnatal heart growth is achieved primarily through hypertrophy of cardiomyocytes, being in line with other findings [29,30,31]. Cessation of the capability of murine hearts to regenerate coincides roughly with the moment that most cardiomyocytes pass through this final stage of DNA synthesis and binucleation [23,32]. Understanding which factors drive the murine postnatal switch from proliferation to terminal differentiation, that is absent in lower vertebrates, may help to find cues to alleviate this restriction.

### 2.3. Human Myocardium Possesses a Restricted Ability for Cardiomyogenesis

Human cardiomyocyte renewal is proven to persist throughout life albeit at a low rate. Bergmann et al. [2] reported that the human myocardium possesses a cardiomyocyte turnover rate of ~2% per year in the first decade of life, to <1% per year by the seventh decade, as estimated by integration of atmospheric 14C into nuclear DNA [2]. Although these numbers rely on mathematical assumptions, and the exact frequency of human cardiomyogenesis awaits validation by other groups and/or methods, these findings show that cardiomyocyte renewal persists throughout human life and derives from pre-existing cardiomyocytes [2]. Furthermore, the possibility of a regenerative window early after birth, like in mice, has been suggested almost a century ago on the basis of post mortem histology [33]. In support, evidence describes functional recovery from corrective heart surgeries in infants [34] and after myocardial infarction of a newborn child [35], suggesting that the human myocardium may also possess a limited timeframe with increased cardiomyogenesis. Taken together, although evidence indicates that cardiomyocyte renewal persists throughout human life and signs of increased cardiomyogenesis shortly after birth have been found, we are still far from understanding its regulation. It is therefore important to identify the molecular basis that limits human adult cardiomyocytes to proliferate. 

### 2.4. Differences and Similarities in the Mechanisms Underlying Cardiomyogenesis between Species

Embryonic mammalian cardiomyocytes are able to divide and share physiological conditions comparable to lower vertebrates [36]. Fish hearts are characterized by a hypoxic environment with relatively lower force generation, while in mammals, the onset of birth demands adaptation to a normoxic environment with the ability to generate high pressure [36]. Cardiomyocytes are able to sense the higher oxygen levels and increased mechanical force and in response remodel themselves and their surroundings to function adequate to the demands of the body [36,37,38]. It is suggested that this postnatal adaptation of the mammalian cardiomyocyte induces fundamental molecular and physiological alterations limiting cellular division. One of the main physiological adaptations is the loss of mononucleated diploid cardiomyocytes, which are suggested to form the only subpopulation to be proliferation-capable [39]. In zebrafish [40] and newt [11], almost all cardiomyocytes remain mononuclear and diploid throughout life. During embryonic mammalian heart development, including rodents [23,30] and humans [41], all cardiomyocytes are mononuclear and diploid. However, only a small number of mononucleated diploid cardiomyocytes persists after birth, accounting for 2.3% to 17.0% in the murine heart depending on the genetic background [39]. Although the exact number of mononucleated diploid cells in the human heart is not known, 30% to 40% of the cardiomyocytes are reported to be binucleated [2,42,43]. The limited fraction of mononuclear diploid cardiomyocytes may mark the boundaries for the regenerative capacity of the mammalian adult heart [3,44,45,46]. 

Surprisingly, despite the discrepancies in myocardial regenerative capacity between lower vertebrates and mammals, the cellular mechanisms controlling cardiomyocyte division seem to be evolutionary conserved. Cell cycle regulators, growth factors, and signaling through the Meis homeobox 1 (Meis1)-, Hippo/Yes-associated protein (YAP)-, switch/sucrose nonfermentable (SWI/SNF)-, neuregulin 1 (NRG1)-, and p38-pathway are proven to exert a fundamental role in cardiomyogenesis during postnatal development [47,48,49]. Furthermore, these factors also control hypertrophic growth of the adult mammalian myocardium in response to pressure overload [50,51,52,53,54,55,56]. These findings suggest that these cellular mechanisms are not only crucial to adapt to the changes during postnatal life but also allow adaptation to cellular stress during adulthood. For further details on a description of these underlying mechanisms, we refer to the current literature available [47,48,49]. 

Overall, comparative analyses of cardiac repair and regeneration in different animals provide important insights into the evolutionary conservation of key processes driving heart regeneration. Although the molecular regulators of cardiomyocyte division remain poorly understood, the similarity in cardiac regeneration capability between lower vertebrates and neonatal mammals is promising as it suggests that humans are able to overcome severe cardiac injury. Increased understanding of the regulators defining the decline in cardiomyocyte proliferation in mammals is required. 

## 3. Noncoding RNAs in Cardiac Regeneration

Non-coding RNA (ncRNA) genes such as tRNA and rRNA genes have long been recognized to play vital roles in the cell, but ncRNA with unknown functionality were initially considered ‘junk DNA’ upon their first identification in 1972 [57]. Nowadays, it is well accepted that they produce functional RNA molecules without a protein coding function, acting as regulators of cellular and tissue function during development and disease [5,58,59]. While the number of identified ncRNA genes is still rising, human gene catalogs already contain more ncRNA genes than protein-coding genes, reaching respectively 25.528 and 19.957 according GENCODE release 33 [60]. NcRNAs can be classified into small (<200 nucleotides) or longer (>200 nucleotides) RNAs [61]. 

MiRNAs are a class of evolutionary conserved small (18–24 nucleotides) ncRNAs, and up to date 2654 microRNAs have been annotated in humans [62]. MiRNAs negatively regulate gene expression of their target genes by inhibiting messenger RNA (mRNA) translation or stability [63,64]. Although the exact mechanism of action of is still not completely clear, the accepted model describes that base pairing of a miRNA with the 5’UTR, protein-coding sequence, or intron of its target mRNA leads to decreased mRNA translation while binding to its 3’UTR results in decay [64]. Target recognition of the miRNA is dictated mainly by perfect pairing of the seed region (nucleotides 2–7 at the 5’end of the miRNA) while partial complementarity of other miRNA regions is also important for specificity and stabilization of the miRNA-mRNA interaction [65]. Single miRNAs are able to interact with up to hundreds of target genes in parallel, while each specific target gene is modestly repressed on transcript [65,66] or protein level [67,68].

Long noncoding RNAs (lncRNA) are genes that do not code for proteins and are defined by an arbitrary criterion of length (>200 nucleotides) [69]. According to NONCODE database (v5), the current number of annotated lncRNA genes in the human genome is 96,308 [70], by far exceeding the number of protein-coding genes (around 20,000–21,000) [71]. This number of lncRNAs is not definitive since novel lncRNA genes are being discovered on a regular basis. LncRNA sequence conservation is limited between species, which does not imply a lack of function but instead suggests that species equipped with lncRNAs display changed (or increased) complexity [72]. This lack of conservation is, partly, compensated for by structural conservation to preserve functionality [72,73]. In contrast to miRNAs, the understanding of the biosynthesis, mechanisms of actions, and functions of lncRNAs is limited. The novelty of the lncRNA research field is also marked by the ongoing progression in defining, categorizing and validating lncRNAs. LncRNAs most commonly are being classified according to their genomic location [59]:Intergenic lncRNAs—located between two protein-coding genes. The majority of lncRNAs are included within this classification.Intronic lncRNAs—located within introns of protein-coding genes.Bidirectional lncRNAs—transcribed within 1 kb of promoters in the opposite direction from the protein-coding transcript.Enhancer lncRNAs—located within close proximity (<2 kb) and transcribed from enhancer regions of the genome.Sense lncRNAs—transcribed from the sense strand of protein-coding genes and can overlap introns and part or all of the exon.Antisense lncRNAs—transcribed from the antisense strand of protein-coding genes and can overlap an exon of the coding gene in the sense strand, an intron, or both.

Although the mechanisms how lncRNAs exert their functions are not yet fully elucidated, a striking difference between miRNAs and lncRNAs is that the latter can either repress or activate gene expression. LncRNAs are mostly localized in the nucleus, where they regulate gene expression at the epigenetic level. LncRNAs make use of very diverse mechanisms to control gene expression, which can be categorized into [74,75]:

Signal lncRNAs—which can regulate gene expression in a time- and space-dependent manner. 

Decoy lncRNAs—which can titrate transcription factors and other proteins away from chromatin or titrate miRNAs out from their target (also known as miRNA sponges). 

Guide lncRNAs—which can recruit chromatin modifying enzymes to target genes, either in cis (near the site of lncRNA production) or in trans to distant target genes. 

Scaffold lncRNAs—which can facilitate the assembly of multiple proteins to form ribonucleoprotein (RNP) complexes, affecting histone modifications on chromatin. Specific microRNAs (miRNAs) and long noncoding RNAs (lncRNAs) have been reported to engage in cardiac regeneration and cardiomyogenesis. Here we review current knowledge concerning the regulatory function of miRNAs and lncRNAs during cardiac regeneration through control of cardiomyogenesis. Understanding and utilizing these master regulators may form the solution to overcome the limited cardiomyogenesis potential of the adult mammalian heart.

### 3.1. MicroRNAs with Widely Studied Regenerative Functions

#### 3.1.1. miR-199a and miR-590

MiRNAs modulate numerous signaling pathways and cellular processes, including cardiac regeneration and cardiomyogenesis. Eulalio et al. [76] performed an unbiased functional screening of 875 human miRNAs for identification of miRNAs possessing the ability to stimulate cardiomyocyte proliferation. Various miRNAs were found to increase DNA synthesis and proliferation in cultured rodent cardiomyocytes. Co-overexpression of miR-199a and miR-590 using adeno-associated virus serotype 9 (AAV9)-mediated gene transfer in murine neonatal hearts increased heart size and enhanced cardiomyocyte proliferation [76]. Importantly, simultaneous overexpression of both miR-199a and miR-590 in murine adult hearts subjected to MI stimulated cardiomyocyte proliferation, cardiac regeneration, and cardiac function [76]. More recently, the same research group made an effort to translate the cardiac regenerative potential of miR-199a towards the clinics by studying its effects in a large animal model [10]. Infarcted pig hearts subjected to AAV6-mediated miR-199a overexpression showed marked improvements in both global and regional contractility, increased muscle mass and reduced scar size in comparison with infarcted, control-treated animals [10]. In line with the rodent studies, miR-199a-overexpressing cardiomyocytes displayed increased DNA synthesis (BrdU incorporation) and elevated expression of proliferation markers, including Ki67 and pH3 [10]. Despite that high frequency of multinucleation in pig hearts [77], replicating cardiomyocytes were mono- or bi-nucleated [10]. However, persistent overexpression of miR-199a resulted in sudden arrhythmic death of most of the treated pigs after approximately 6 weeks. Such events were concurrent with myocardial infiltration of proliferating cells displaying a poorly differentiated myoblastic phenotype [10].

#### 3.1.2. miR-15 Family

Profiling studies during postnatal cardiac development identified upregulation of multiple members of the miR-15 family when comparing the onset of cardiomyocyte cell cycle arrest and binucleation (postnatal day 10) with highly proliferative state (postnatal day 1) [6]. These family members, including miR-195, miR-15a, miR-15b, miR-16 and miR-497, share the same seed sequence and therefore target similar genes and control similar pathways. In mice, gain- and loss-of-function studies show that miR-15 family members inhibit cardiomyocyte proliferation during development and in response to cardiac injury. MiR-195 was the most upregulated miRNA when comparing hearts deriving from 1-day old mice in comparison with 10-day old ones [6]. Preliminary overexpression of miR-195 in embryonic murine hearts leads to ventricular hypoplasia and ventricular septal defects and diminished cardiomyocyte proliferation [6]. Transgenic overexpression of miR-195 reduced cardiomyocyte proliferation and the regenerative capacity of 1-day old mice in response to MI [78]. Consistently, inhibition of the miR-15 family prior infarction in mice through administration of miR-15 antimiRs, reduced infarct size and preserved cardiac function in response to ischemia-reperfusion injury [78,79]. The beneficial effect of inhibiting miR-15 family members observed in mice was associated with an increased number of mitotic cardiomyocytes [79]. Target genes of the miR-15 family explaining its proliferation-inhibiting function include Check1 (checkpoint kinase 1) amongst others [6]. 

An additional miR-15 family member identified to be induced during postnatal development and contributing to cardiomyocyte cell cycle arrest is miR-128. In neonatal mice, cardiac-specific overexpression of miR-128 impaired cardiomyocyte proliferation and cardiac function, while miR-128 deletion extended proliferation of postnatal cardiomyocytes [80]. Furthermore, deletion of miR-128 promoted cell cycle re-entry of adult cardiomyocytes, reduced cardiac fibrosis, and attenuated cardiac dysfunction in response to MI [80].

#### 3.1.3. The miR-1/-133 Cluster

MiRNA genes can be located within close genomic proximity, so-called miRNA clusters. Evolutionary highly conserved miRNAs are more frequently organized in clusters [75]. It is proposed that miRNAs belonging to the same cluster target overlapping sets of genes and therefore share similar functions [75]. Indeed, this has also been proven for specific miRNA clusters with a functional involvement during cardiac regeneration through regulation of cardiomyogenesis.

Among the most abundantly expressed miRNAs in the myocardium is the highly conserved miR-1/miR-133 cluster, which has been shown to play a fundamental role in cardiomyogenesis and cardiac development by different groups. Both miRNAs exhibit cardiac- and skeletal muscle-enriched expression [81,82]. MiR-1 is the most abundant miRNA in the mammalian heart, accounting for almost 40% of all miRNA expression in the adult murine heart [83]. MiR-1 inhibits cardiomyocyte proliferation through, at least partly, targeting of Hand2 [81]. Transgenic miR-1 overexpressing mice display a disrupted embryonic heart development associated with thin-walled ventricles, heart failure, and lethality at E13.5 [81]. Transgenic mice lacking miR-1 have an aberrant cardiomyogenesis, cardiac conduction, ventricular hypoplasia, and cell cycle progression [84]. These effects indicate that miR-1 exerts a crucial and time-dependent function in controlling cardiomyocyte proliferation during development.

Similarly, miR-133 negatively regulates cardiomyocyte proliferation by inhibiting Cyclin D2 and Serum Response Factor [85]. Expression of miR-133 has been shown to be downregulated in regenerating zebrafish heart [86]. Functional modulation of miR-133 has been shown to influence the regenerative capacity of the zebrafish myocardium, in which forced expression diminishes its regenerative potential while miRNA sponge-mediated miRNA inhibition promotes regeneration [86]. Knock-out mice lacking either miR-133a-1 or miR-133a-2 do not display phenotypical alterations, while deletion of both miRNA genes results in ventricular-septal defects and embryonic lethality in 50% of the mice. The surviving double knock-out mice display dilated cardiomyopathy and heart failure [85]. On the contrary, gain-of-function of miR-133 in embryonic cardiomyocytes diminished cardiomyocyte proliferation and resulted in ventricular wall thinning and increased lethality [85]. Although miR-1 and miR-133 have different seed regions and therefore target a different set of genes, their function in regulating cardiomyogenesis is highly similar. These studies mark the miR-1/miR-133 cluster as a key regulator of cardiomyogenesis during development and in response to injury.

#### 3.1.4. The miR-302-367 Cluster

The miR-302-367 cluster has been shown to be essential for cardiomyocyte proliferation and cardiac repair in mice. Expression of the miR-302-367 cluster is present in the embryonic mouse heart and diminishes during maturation [87]. Re-expression of this cluster suffices to reactivate cardiomyocyte cell cycle progression and proliferation during development and in response to MI. However, prolonged forced expression of miR-302-367 in mice through intravenous administration of miRNA mimic induced cardiomyocyte dedifferentiation and cardiac dysfunction, suggesting that persistent reactivation in postnatal cardiomyocytes is not desirable [87]. Transient systemic application of miR-302-367 mimics overcomes this problem and was shown to increase cardiomyocyte proliferation, decrease fibrosis, and improve function after MI in mice [87]. The miR-302-367-mediated induction of cardiomyocyte renewal and proliferation is caused, at least in part, by inhibition of the Hippo pathway by direct targeting of Macrophage Stimulating 1 (Mst1), Large tumor suppressor 2 (Lats2), and MOB Kinase Activator 1B (Mob1b) [87].

#### 3.1.5. The miR-17-92 Cluster

The miR-17-92 is among the best-studied miRNA clusters and was initially identified as a human oncogene that promotes cell proliferation of different types of tumors through phosphatase and tensin homolog (PTEN) targeting [88]. Deletion of miR-17–92 cluster from embryonic and postnatal mouse hearts reduced cardiomyocyte proliferation and cardiac development, while transgenic overexpression is sufficient to induce cardiomyocyte proliferation in embryonic, postnatal, and adult hearts [89]. Furthermore, forced expression of this miRNA cluster in adult cardiomyocytes protects against MI-induced scar formation and cardiac dysfunction [89].

#### 3.1.6. The Cluster of miR-99/100 and Let-7 Families

Aguirre et al. [7] identified ~60 miRNAs to be differentially regulated in zebrafish three days post amputation of the ventricular apex in comparison with uninjured hearts. Two evolutionary conserved microRNA families (miR-99/100 and Let-7a/c), which are located in miRNA clusters on two genomic locations (miR-99/Let-7c and miR-100/Let-7a) are strongly downregulated during regeneration [7]. Administration of miR-99/100 miRNA mimics impaired the regenerative capacity of the adult zebrafish heart in response to ventricular resection, while miRNA antagomiRs promoted it [7]. The murine heart also displayed increased recovery from MI upon AAV-mediated gene transfer of antimiRs against miRNA-99/100 and Let-7a/c [7]. It is important to note that these clusters do not only consist of miR-99/100 and Let-7 family members but also members of the miR-125 family [90]. The putative involvement of the miR-125 family in cardiomyocyte proliferation has not been functionally studied yet, although miR-125a/b/c have been identified to be differentially expressed during murine postnatal development [6] and crucially involved in proliferation of various tumors (Reviewed in [91,92,93]).

In addition to the well-studied miRNA clusters and families, multiple other miRNAs have been reported to contribute to cardiac repair through regulating cardiomyogenesis, including miR-222 [94], miR-31a [95], miR-294 [96], miR-204 [97], and miR-29a [98].

#### 3.1.7. Other miRNAs Implicated in Cardiomyocyte Cell Cycle Regulation

In addition to the well-studied miRNA clusters and families, multiple other miRNAs have been reported to contribute to cardiac repair through regulating cardiomyogenesis, including miR-222 [94], miR-31a [95], miR-294 [96], miR-204 [97], and miR-29a [98]. Transgenic mice overexpressing miR-222 in their hearts display improved cardiac contractility and a 70% reduction in scar formation in response to ischemia-reperfusion injury in comparison to wild-type mice [94]. Xiao et al. [95] identified miR-31a-5p levels to be upregulated in cardiomyocytes deriving from 10-day old mice in comparison with 1-day old mice. MiR-31a-5p promoted proliferation markers in cultured cardiomyocytes upon forced expression, while miRNA inhibition had opposite effects [95]. MiRNA knockdown in neonatal mice through antagomiR treatment at day 0 for 3 consecutive days reduced cardiomyocyte proliferation markers [95]. More recently, Borden et al. [96] identified miR-294 to be expressed in the prenatal- but not in the neonatal or adult heart. AAV9-mediated gene transfer of miR-294 to murine hearts for 14 days continuously after MI significantly increased myocyte cell cycle reentry and left ventricular functions together with decreased infarct size and apoptosis [96]. MiRNA-204 has been shown to promote both neonatal and adult cardiomyocyte proliferation in vitro [97]. Cardiac-specific overexpression of miRNA-204 in transgenic mice exhibits excessive cardiomyocyte proliferation in the embryonic and adult stages, indicated by increased expression of various cell cycle regulators and increased myocardial mass [97]. Cao and colleagues [98] identified miR-29a as one of the most highly upregulated miRNAs when comparing the rat myocardial miRNA profile of 2 days with 4 weeks after birth [98]. MiR-29 increased, while its inhibition reduced cell cycle activity of cultured neonatal cardiomyocytes, marked by an altered expression of proliferation markers Ki-67, Cyclin D2, and Cyclin-dependent kinase 2 [98].

Taken together, evidence accumulating over the past two decades marks the crucial regulatory role of various miRNA clusters and families in cardiac regeneration through controlling cardiomyogenesis (Table 1). MiRNA-based biotherapeutics have been shown to be successful in boosting the regenerative capacity of the injured adult mammalian heart in small and large animal models. These approaches hold great promise for validation and await further investigation.

### 3.2. Long Noncoding RNAs

LncRNAs in general are considered to be expressed at a relatively low level [99]. However, various specific lncRNAs display a cardiac-specific, or at least cardiac-enriched expression signature [100,101]. Reported numbers of lncRNAs expressed in the heart vary from approximately ~100 [100] to ~2500 [101] depending on the sequencing technique and applied lncRNA annotation pipeline. Moreover, profiling studies reported alterations in lncRNA expression levels during cardiac development and disease in animals and humans [101,102,103,104]. However, characterization of the functional roles of these candidate lncRNAs in the developing or infarcted myocardium remains a challenging task.

#### 3.2.1. CARMEN

Ounzain et al. [8] characterized the lncRNA transcriptome during cardiac differentiation of progenitor cells obtained from the human fetal heart and identified 570 lncRNAs that were modulated during cardiomyocyte differentiation [8]. Many of these were associated with active cardiac enhancers as their expression correlated with that of proximal cardiac genes. Among the most upregulated lncRNAs was an enhancer-associated lncRNA named CARdiac Msoderm Enhancer-associated Noncoding RNA (CARMEN). The authors demonstrated that CARMEN knockdown inhibits cardiac specification and differentiation in cardiac precursor cells independently of miR-143 and -145 expression, two microRNAs located proximal to the enhancer sequences [8]. CARMEN has the ability to epigenetically modulate the expression of important cardiac transcript factors and is therefore a crucial regulator of cardiac cell differentiation and homeostasis [8].

#### 3.2.2. ANRIL

Within the human genome, a crucial cell cycle-modulating genomic locus spans approximately 42 kbs on human chromosome 9p21. This region contains three important tumor suppressors: p15, p16, and p14 [105,106]. Antisense to p15 and p16 is located the 3.8 kb-long ncRNA ANRIL (antisense noncoding RNA in the INK4 locus) [107]. ANRIL expression correlated with atherosclerosis severity and increased cellular proliferation [107]. RNA immunoprecipitation demonstrated that deletion or knock-down of ANRIL removed the epigenetic restriction on the p15 locus but not on p14 or p16, increasing its expression while decreasing cell cycle activity [105,106,108]. At the molecular level, ANRIL binds to epigenetic effector proteins PRC1 and PRC2 and recruits them to promoter regions to regulate gene expression patterns [105,106,108]. Thus, ANRIL is a lncRNA participating in epigenetic transcriptional control of proliferation-associated genes. Whether or not cardiomyocyte proliferation depends on ANRIL-mediated epigenetic control still needs to be determined. 

#### 3.2.3. Braveheart

Klattenhoff and colleagues [109] identified Braveheart (Bvht) as a heart-associated lncRNA, highly induced during and required for prenatal cardiac development [109]. This lncRNA is ~590 nucleotides long and located on the positive strand of mouse chromosome 18 while not being detectable in human or rat hearts. Deletion of Bvht impairs embryonic stem cell differentiation into cardiomyocytes and plays a role in maintaining cardiac fate in neonatal cardiomyocytes [109,110]. Bvht drives cardiac commitment through epigenetic control of a core cardiovascular gene network during cardiomyocyte differentiation [109]. More recently, the same research group showed that the RNA secondary structure is key to exert this role and that a 5’ asymmetric G-rich internal loop is necessary for cardiomyocyte differentiation [110]. 

#### 3.2.4. Fendrr

Grote et al. [9] identified FOXF1 adjacent noncoding developmental regulatory RNA (Fendrr) to be highly expressed in embryonic mice in the posterior mesoderm (which gives rise to the heart) [9]. This antisense lncRNA is 3099 nucleotides long and transcribed bidirectionally with FOXF1 [111]. Transgenic Fendrr knock out mice displayed increased embryonic mortality associated with ventricular defects, possibly as a result of impaired cardiomyocyte proliferation [9]. Like Bvht, Fendrr acts as an epigenetic regulator of chromatin structure to shape the gene expression profile during cardiac commitment [9]. The authors concluded Fendrr to be essential for proper heart and body wall development in the mouse [9]. More recently, Sauvageau et al. [112] also created a Fendrr knockout mouse model using a different genetic approach, resulting in ventricular heart defects at a comparable embryonic stage [112]. These findings indicate that Fendrr is crucially involved in cardiac development. Whether Fendrr only plays a role in cardiomyocyte commitment or also cardiac regeneration through cardiomyogenesis still needs to be investigated. 

#### 3.2.5. H19

The expression of lncRNA H19 is regulated during development and implicated in cancer [113]. H19 harbors multiple binding sites complementary to Let-7 miRNA family members [7]. Crosslinking experiments upon H19 modulation prove that this lncRNA acts as a molecular sponge to prevent Let-7 family members from inhibiting their target genes [113,114]. As discussed before, the Let-7 family itself is known to be crucially involved in development, cancer, and cardiomyocyte proliferation [7,115]. In line, down-regulation of H19 has been shown to promote the differentiation potential of mouse embryonic stem cells into cardiomyocytes [114]. These findings identify this lncRNA as an important regulator of the Let-7 family of miRNAs and can therefore potentially play a crucial role in cardiomyogenesis. 

#### 3.2.6. Dlk1-Dio3

The murine Delta-like homolog 1 (Dlk1)- type III iodothyronine deiodinase (Dio3) ncRNA locus is a mega-cluster (~1.000 kb) consisting of 3 protein coding genes, 61 miRNAs, and 3 lncRNAs [116,117]. The organization of the locus and most of its harboring ncRNAs are conserved between mammals [118]. It is not completely understood how expression of the numerous ncRNAs deriving from the Dlk1-Dio3 locus is controlled. Several reports suggest that Dlk1-Dio3-deriving ncRNAs are co-transcribed and subsequently undergo post-transcriptional processing into individual mature ncRNAs [119,120,121]. Other evidence indicates that expression of various ncRNAs within the locus are under the control of distinct enhancers [122,123]. Regardless of its regulation, humans with abnormalities in the Dlk1-Dio3 locus are characterized by impaired fetal development and postnatal growth [124,125,126], of which a small number of patients suffer from cardiac abnormalities [127,128]. In support, expression of specific Dlk1-Dio3-deriving miRNAs, such as miR-410 and miR-495, has been reported to decrease in the mammalian heart during the course of development [119,129,130]. Although it is challenging to functionally study the Dlk1-Dio3 locus as a whole, the screening of Eulalio et al. [76] identified individual Dlk1-Dio3-deriving miRNAs as stimulators of proliferation in neonatal rat cardiomyocytes, including miR-154, miR-380, miR-410, miR-411, miR-495, miR-539, miR-668 [76]. In addition, Clark et al. [129] have confirmed the stimulating function of miR-410 and miR-495 on cardiomyogenesis in cultured neonatal rat cardiomyocytes. In contrast to the miRNAs, the lncRNAs deriving from the Dlk1-Dio3 locus (Gtl2, AntiRtl1, and Mirg) have not been studied for their role in cardiomyogenesis yet.

Thus, over the past few years an increasing number of profiling studies identified lncRNAs to be cardiac-enriched and/or differentially expressed during myocardial development or disease (Table 2). Although these heart-associated lncRNAs have emerged as important regulators in cardiac development and disease, their exact functionality and therapeutic potential remain to be determined. Moreover, miRNAs and lncRNAs may affect the same pathways and interact with each other. Further research is needed to elucidate the complete network of miRNAs, lncRNAs and mRNAs involved in controlling cardiomyocyte proliferation. 

## 4. Current Limitations and Promises to Control Cardiac Regeneration through Modulation of Noncoding RNAs

Unraveling the cellular process of cardiomyocyte proliferation and the regulatory networks involving ncRNAs has yielded promising candidates to trigger cardiac regeneration and repair. Although huge progress has been made, major challenges have to be overcome in order to increase our understanding and to ultimately utilize this knowledge to boost repair of the adult mammalian heart.

### 4.1. Translating Cardiomyogenesis-Stimulating Therapies towards Larger Animals

First and most importantly, cardiomyocyte division in the adult mammalian heart is an extremely rare event. A tremendous number of cardiomyocytes are lost during MI. Repairing the heart by generating a sufficient number of cardiomyocytes to compensate for the tissue loss and to preserve function is challenging. To date, ncRNA-based therapeutic approaches that are successful in promoting cardiomyogenesis are predominantly performed in small animal models, with only few studies making use of large animals [10,132]. Whether the mechanism underlying the successful approaches in small animals are preserved in large animals or eventually humans remains to be demonstrated. Sequence and structure conservation of ncRNA and their interaction partners plays a fundamental role in this. Besides, applied therapies need to function under different physiological conditions, such as an increased pressure and decreased frequency of proliferation competent cells. As discussed, it has been suggested that only the subpopulation of mononuclear diploid cardiomyocytes is competent to undergo proliferation [39]. Repairing the human injured heart through division of only this small fraction will therefore not suffice. Increasing the regenerative capacity of the human heart may necessitate a higher fraction of proliferation-competent cardiomyocytes. 

### 4.2. Quantification of True Cardiomyocyte Division Is Challenging

Second, measuring true cardiomyocyte division remains a technical challenge because of biological and technical factors. Cellular count of cardiomyocytes isolated via tissue dissociation are often subject to inconsistent results and selection bias due to variation in digestion efficiency. Similarly, whole heart nuclear fraction isolation followed by detection of cardiomyocyte specific nuclei using PCM-1 has been performed [133]. However, this approach does not correct for cardiomyocyte binucleation. To date, more direct proof of elevated cardiomyocyte number in the heart can be determined using stereology. Stereology calculates the total number of cardiomyocytes by measuring cardiomyocyte number and size in successive tissue sections [28,131]. Another useful alternative approach is clonal analysis. Multiple genetic mouse models have been developed to evaluate cardiomyocyte proliferation, including the FUCCI- [134], TRE-YAP- [135], and MADM-models [136]. Described stereology measurements and genetic mouse models are however costly and restricted to centers of expertise. 

Monitoring of various cell cycle activity markers in cardiomyocytes is therefore more commonly used as indicator of proliferation. Determination of these indirect surrogate markers implicated in proliferation can be done in a ‘static’ or ‘dynamic’ approach. The latter is applied when detecting cardiomyocyte DNA synthesis over time by nucleotide analog (EdU or BrdU) incorporation in cultured cardiomyocytes or in vivo. Static evaluation of cardiomyocyte proliferation markers is often performed by quantification of the percentage of cardiomyocytes positive for the cell cycle activity marker Ki67, the mitotic marker phosphohistone H3, or Aurora kinase B, which is expressed during binucleation and division. Although these markers are active during all or specific cell cycle phases, their expression does not necessarily indicate definitive cardiomyocyte division. Therefore, Milliron et al. [137] very recently developed a technique in which fluorescence-activated cell sorting for Cdc20 and Spg20 distinguishes between diploid, binucleated, and polyploid human induced pluripotent stem cell-derived cardiomyocyte populations [137]. While each of the previous markers is helpful for building evidence that cardiomyocytes are cell cycle active, they should be accompanied with more direct proof of cardiomyocyte proliferation. 

### 4.3. Large Scale Identification of ncRNAs Associated with Cardiomyogenesis and Cardiac Regeneration

Large scale identification of ncRNAs associated with cardiomyogenesis improved significantly with the advancements in genome-wide and RNA deep-sequencing techniques. Various groups have performed miRNA and/or lncRNA profiling of models associated with increased cardiomyogenesis, including the postnatal switch from proliferation to terminal differentiation [6,7,8]. Although this approach has shown to bring forward promising ncRNA candidates, alternative approaches may lead to a more sensitive and specific identification. For miRNAs particularly, it has been shown that differential expression does not necessarily imply functionality; miRNA association with the RISC complex is a superior predictor of functionality than miRNA expression levels [138]. Determination of RISC loading in models of increased cardiomyogenesis, being the developing or injured heart, could potentially lead to novel miRNA candidates. Furthermore, lncRNA research, specifically, is restricted by the fact that the majority of lncRNAs are poorly conserved at the sequence level among different species, necessitating the use of humanized models or patient data for translation. 

Genome-wide association studies (GWAS) are used to identify genomic regions associated with a disease phenotype by scanning along the genome for known single-nucleotide polymorphisms. Indeed, GWAS have associated a plethora of SNPs among the ~20000–21000 protein-coding genes with cardiac disease [139,140]. SNPs may affect the expression, tertiary structure, or interaction of genes, regardless of their coding potential. Remarkably, Meurs et al. [141] identified SNPs located in the 3’UTR of protein coding genes that interfere miRNA-mRNA interaction [141]. Strikingly, up to ~88% of SNPs identified in GWAS are located in noncoding regions [142], but only few are being further pursued [107,108,143]. Applying GWAS to associate genomic regions among the 2654 miRNA- and 96,308 lncRNA genes in the human genome with cardiomyogenesis may hold great promise.

### 4.4. NcRNAs Function Needs to Be Controlled at the Right Time, in an Optimal Dose, and in a Specific Cell Type

Another hinderance to overcome when embarking on finding therapeutic strategies for cardiac regeneration is to efficiently modulate ncRNA function, in a time-, dose-, and cell type-specific manner. Following identification of a cardiomyogenesis-associated ncRNA, gain- and loss-of-function studies need to determine its functional involvement. Transgenic animal models can be generated to cell- and/or time-specifically modulate the expression of a ncRNA in an all-or-nothing approach, but this can be costly and time-consuming. Sophisticated tools to better control ncRNA function have been well established for miRNAs, although the therapeutic options for lncRNAs are more limited due to their greater length [144]. NcRNA knockdown or overexpression can be achieved in vitro and in vivo by administration of inhibitors or mimics, respectively. These therapeutic approaches allow for safe and precise control of dosing and time in small [145] and large [146] animal models as well in patients [147]. However, systemic administration of these particles results in an altered candidate function throughout the body. Importantly, uptake of systemically administered inhibitors or mimics by cardiomyocytes is rather inefficient, demanding increased dosing with potential toxic effects as a result [144]. Local delivery into the heart successfully alters ncRNA function at relatively lower dose but can be invasive [144,148]. 

The recent advancements in AAV-mediated gene therapy allow for prolonged cardiomyocyte-enriched delivery of transcripts (with a maximum length of ~4.7kb) at a specific moment during development or disease (Reviewed in [149,150]). Gene transfer using such vectors of protein-coding genes as well as miRNA mimics and inhibitors have been utilized to interrogate their involvement in cardiac regeneration in various animals [148]. In humans, the first AAV-mediated cardiac gene transfer has been shown to be safe in Phase IIb clinical trials in patients suffering from heart failure [151]. However, the clinical trial failed to meet its primary end point, potentially explained by low transduction of myocardial cells by AAV1 [151]. Since then, vast improvements have been made to more efficiently and specifically transduce cardiomyocytes through optimizing the AAV virus capsid and serotype [152,153]. As a result, AAV9 is currently the most commonly used serotype to transduce the hearts of small and large animals. Notably, the first Phase I clinical studies making use of AAV9-mediated gene transfer to treat cardiac disease are initiated and monitored with high expectations (clinicaltrials.gov NCT03882437 and NCT02240407). This revolutionary research field is making vast progression and holds great promise to provide reliable, safe, and prolonged treatment options for future cardiac ncRNA-based biotherapeutics.

### 4.5. Deciphering the Complex Network of ncRNAs and Their Interaction Partners

Finally, once ncRNAs associated with cardiac regeneration have been identified, the complex network of ncRNAs and their interaction partners has to be elucidated. A commonly used tool to study interaction between miRNAs and their targets is a luciferase assay. This assay investigates the interaction between miRNAs and their targets, but often only (part of) the target gene’s 3’UTR is incorporated. As a result, the assay may suffer from a lack of preservation of the tertiary structure of the studied RNA molecules. Moreover, luciferase assays performed in non-cardiomyocyte cells may not offer a representative intracellular transcriptome and therefore fail to identify relevant transcripts [138,154]. 

RNA and DNA immunoprecipitation assays give a more biologically relevant representation of interactions between miRNAs, lncRNAs, mRNAs, and proteins. Recent developments in immunoprecipitation methods using Argonaute 2—the main interacting protein that provides miRNAs their inhibitory function- enables largescale identification of miRNA-mRNA networks in a cardiac-relevant in vitro [155] and in vivo [156] setting. Interestingly, Spengler and colleagues [157] have performed crosslinking immunoprecipitation of Argonaute 2 with miRNAs and mRNAs, coupled with high-throughput sequencing (HITS-CLIP). This resulted in the first transcriptome-wide map of miRNA targeting events in the human myocardium, detecting 4000 cardiac Ago2 binding sites across >2200 target transcripts [157]. Applying an identical approach in models of increased cardiomyogenesis, such as injured hearts deriving from zebrafish or mouse neonates, will result in a rich source of miRNAs and their target genes controlling cardiac regeneration. Using these tools to define the ncRNA-RNA interactome would therefore help to unravel the regulatory network of ncRNAs and their interaction partners in the process of cardiac regeneration.

## 5. Conclusions

Regeneration of the adult mammalian heart is limited, but recent findings have marked its capacity to recover from severe injury early after birth, and to renew cardiomyocytes throughout life. Monitoring the regenerative capacity of the heart along evolution has taught us that several many underlying cellular mechanisms may be shared between lower vertebrates and neonatal mammals. This holds promise as it suggests that cardiomyogenesis-regulating ncRNAs identified in animal models may act in a similar manner in humans.

Accumulating evidence has made it clear that both short and long ncRNAs are important orchestrators of gene regulatory networks in this process. A remarkable amount of evolutionary conserved miRNA clusters and families have been proven to control cardiomyocyte division, including the miRNA cluster of miR-99/100 and Let-7 families. Similarly, an increasing number of studies demonstrate that cardiomyocyte proliferation is under epigenetic control of the more recently emerged class of lncRNAs, such as ANRIL. MiRNA research kicked off more than two decades ago and is therefore several steps ahead in comparison with the field of other classes of ncRNAs. This may explain the superior body of knowledge of miRNAs in cardiac regeneration over lncRNAs. At the same time, this also provides hope for future developments since significant advancements have been achieved by studying 2654 miRNAs, while the 96,308 lncRNA genes forms an immense, unexplored source to explain the limitations of cardiac regeneration. Recent advancements in sequencing and gene therapy technologies improved largescale identification and precise in vivo modulation of ncRNAs to further elucidate the regulatory networks involved in cell cycle arrest and activation. Understanding and employing these critical regulators of cardiomyocyte proliferation, especially the miRNAs and lncRNAs, will aid our search to remove the breaks on myocardial regeneration.

## Figures and Tables

**Table 1 biomolecules-10-00262-t001:** MicroRNAs regulating cardiac regeneration through control of cardiomyogenesis.

ncRNA	Model	Disease	Therapy	Observed Effect	Reference
miR-199a and miR-590	Neonatal mouse	-	AAV9-mediated OE	CM proliferation ↑, heart size ↑	[76]
	Adult mouse	MI	AAV9-mediated OE	CM proliferation ↑, regeneration ↑, function ↑	[76]
miR-199a	Pig	MI	AAV6-mediated OE	CM proliferation ↑, contractility ↑, muscle mass ↑, scar size ↓, sudden arrhythmic death	[10]
miR-195	Embryonic mouse	-	Transgenic cardiac OE	Heart size ↓, CM proliferation ↓, ventricular defects	[6]
	Neonatal mouse	MI	Transgenic cardiac OE	CM proliferation ↓, regeneration ↓, ventricular defects	[78]
miR-15 family	Neonatal mouse	-	AntimiR-mediated INH	CM proliferation ↑	[6]
	Neonatal/adult mouse	MI	AntimiR-mediated INH	CM proliferation ↑, function ↑, regeneration ↑	[78]
	Adult mouse	MI	AntimiR-mediated INH	CM proliferation ↑, Infarct size ↓, function ↑, regeneration ↑	[79]
miR-128	Neonatal mouse	-	Transgenic cardiac OE	CM proliferation ↓, function ↓	[80]
	Neonatal mouse	-	Transgenic cardiac KO	CM proliferation ↑,	[80]
	Adult mouse	MI	Transgenic cardiac KO	CM proliferation ↑, Infarct size ↓, function ↑	[80]
miR-1	Embryonic mouse	-	Transgenic OE	Ventricular defects, mortality ↑	[81]
	Embryonic mouse	-	Transgenic OE	CM proliferation ↓, ventricular defects	[84]
miR-133	Zebrafish	VR	Transgenic OE	Regeneration ↓	[86]
	Zebrafish	VR	Sponge-mediated INH	Regeneration ↑	[86]
	Embryonic mouse	-	Transgenic double KO	Ventricular defects, heart failure, mortality ↑	[85]
	Embryonic mouse	-	Transgenic OE	CM proliferation ↓, mortality ↑	[85]
miR-302-367	Neonatal mouse	-	Transgenic OE	CM proliferation ↑	[87]
	Adult mouse	-	Transgenic cardiac OE	CM proliferation ↑, function ↓	[87]
	Adult mouse	MI	Mimic-mediated OE	CM proliferation ↑, Infarct size ↓, function ↑	[87]
miR-17-92 cluster	Embryonic/neonatal mouse	-	Transgenic cardiac KO	CM proliferation ↓, ventricular defects	[89]
	Embryonic/neonatal mouse	-	Transgenic cardiac OE	CM proliferation ↑	[89]
	Adult mouse	MI	Transgenic cardiac OE	CM proliferation ↑, heart size ↑, infarct size ↓, function ↑	[89]
miR-99/100-Let-7 cluster	Zebrafish	VR	Mimic-mediated OE	CM proliferation ↓, regeneration ↓	[7]
	Zebrafish	VR	AntimiR-mediated INH	CM proliferation ↑, regeneration ↑	[7]
	Mouse	MI	AAV-antimiR INH	CM proliferation ↑, regeneration ↑, function ↑	[7]

AAV, Adeno-associated virus; CM, Cardiomyocyte; INH, Inhibition; KO, Knock-out; MI, Myocardial Infarction; OE, Overexpression; VR, Ventricular resection.

**Table 2 biomolecules-10-00262-t002:** Long noncoding RNAs regulating cardiac regeneration through control of cardiomyogenesis.

ncRNA	Model	Therapy	Observed Effect	Reference
CARMEN	P19CL-6	shRNA-mediated INH	CM differentiation ↓, CM proliferation ↓	[8]
ANRIL	HEK293	Stable cell line	Cell adhesion ↑, growth ↑, metabolic activity ↑, apoptosis ↓	[107]
Braveheart	mESC	shRNA-mediated INH	CM differentiation ↓	[131]
Braveheart	nRCM	shRNA-mediated INH	CM size ↓, differentiation ↓	[131]
Fendrr	Embryonic mouse	Transgenic KO	CM proliferation ↓, ventricular defects	[9,112]
H19	P-ESC	shRNA-mediated INH	CM differentiation ↓	[114]

ShRNA, Short hairpin; CM, Cardiomyocyte; INH, inhibition; KO, Knock-out; mESC, Mouse embryonic stem cell P-ESC, Parthenogenetic embryonic stem cell; nRCM, Neonatal Rat Cardiac Myocyte.

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
