# Peer review of "Reviewing the Limitations of Adult Mammalian Cardiac Regeneration: Noncoding RNAs as Regulators of Cardiomyogenesis"

_biomolecules, 2020, doi:10.3390/biom10020262_

Round 1

Reviewer 1 Report

“Reviewing the Limitations of Adult Mammalian Cardiac Regeneration: Noncoding RNAs as Regulators of Cardiomyogenesis” by Verjans et al. summarizes evidence that numerous noncoding RNAs (ncRNAs) such as miRNAs, miRNA families/clusters and lncRNAs are important regulatory factors in cardiomyocyte biology - namely proliferation, differentiation, and regeneration. This review highlights information on cardiac regeneration in several model systems. In addition, because of the central importance of regulating cardiomyocyte transitions between proliferation and differentiation the roles of some of the ncRNAs in these cellular processes are also discussed. Finally, two tables are provided that clearly and succinctly summarize the observed phenotypic effects of various noncoding RNAs on cardiomyocytes, which effectively complements the information provided in the text. The authors do a commendable job linking the various ncRNAs to cardiac proliferation, differentiation, and regeneration. The information listed in the tables is clearly depicted and to the point, without extraneous information. Because the authors highlight a section focusing on “Current limitations and promises to control cardiac regeneration…” it would be worthwhile to also include additional miRNAs that have actually been directly demonstrated to control proliferation of cardiomyocytes (see comment below) even in in vitro model systems. This information is a step ahead functionally than some of the large scale screens mentioned in section 4.3 and their documented role in proliferation could be highly relevant to study in the context of regeneration. Overall, this is a well written review paper containing pertinent information on miRNAs and lncRNAs in cardiac regeneration and cardiomyogenesis.

Major Concern:

While the authors do a commendable job on highlighting well established miRNAs and lncRNAs in cardiomyocyte biology it is surprising the authors failed to cite an emerging noncoding cluster, Dlk1-Dio3 (or Gtl2-Dio3), and its role in cardiomyocyte proliferation (miR-410, miR-495, and others in the cluster), which has implications in cardiac regeneration. Given the importance of regulating proliferations the author ought to include these findings.

Reviewer 2 Report

The paper by Verjans and colleagues provides an overview of the literature of the functions of non-coding RNA in cardiomyogenesis and other aspects of cardiac regeneration. The paper is mostly well written and covers the necessary topics and so I am happy to endorse its acceptance for publication if the following minor points are addressed.

Abstract. Line 15 replace “human” with “mammalian” Abstract. Line 15 insert “limited” between “undergoes” and “cardiomyocyte” In a few different sections of the manuscript, authors refer to “master regulators” but I think this notion is over-simplistic. We know that proteins work in complexes, pathways and networks and it is likely that ncRNAs are the same. My suggestion would be to replace the term “master regulators” to either “regulatory networks” or just “regulators” throughout the paper. P 2 L 45 replace “revolutionary” with “ambitious” P 3 L 97 replace “is set” with “are set” P3 L 113 Spelling “mathods” P3 L 114 Delete “and” at the end of the sentence P3 L 115 Replace “has been suggested for almost a century” to “has been suggested almost a century ago on the basis of post mortem histology” P 4 L 142 Spelling “discrepancies” P 4 L 148 Spelling “growth” P 4 L 152-154 This passage is speculative and should be deleted. P 4 L163 This part is problematic. The quote of 98% of the genome being transcribed into ncRNA is obtained from reference 57, but in that paper, the 98% figure is referenced to the original mouse genome paper [PMID: 12466850]. In that paper, the 98% figure is nowhere to be seen. My recommendation here is to use a newer estimate of the size of the non-coding portion of the genome based on GENCODE (https://doi.org/10.1093/nar/gky955). In addition, the passage “Although ncRNAs were formerly considered as ‘junk DNA’ ...” is totally false. ncRNAs such as tRNA and rRNA perform vital roles in the cell. This section would be better served by first recognising that some ncRNAs like tRNAs and rRNAs are vital, but these are not the focus of this review.  P 4 L 168 in genetics you can use the abbreviation “nt” for nucleotides P 4 L 173 please replace “dogma” with “model”. (Dogmas have no place in science) P 4 L 184 Thousands separator should be “,” P 4 L 185 I suggest here adding a passage to discuss whether lncRNAs are expressed in specific cell types/niches or have broad low level expression.  P 5 L 205 “Although the molecular mechanisms underlying lncRNAs are not yet fully elucidated” This phrase is problematic. Do you mean the way these lncRNAs regulate downstream genes? P 5 L 210 “Signal” should be bold. P 5 L 211 Add in parentheses that decoy RNAs are also known as RNA sponges. P 5 L 216 Add a period at the end of the sentence. P 5 L 210 to 215 Add some detail about circular RNAs here (eg: https://doi.org/10.1038/s41569-019-0185-2) P 6 L 281 Avoid vague words like “dysregulation” whenever possible. Replace with “impaired” or “aberrant” as necessary. P 7 L 310 “Hippo” is capitalised P 9 L 370 Replace “till” with “to” P 9 L 371 Delete “used” P 9 L 389 Put a whitespace between the numeral and the unit. Eg: 42 kbp. P 11 L 468 Replace “taunted by” with “subject to”
